# MINEDOJO: Building Open-Ended Embodied Agents with Internet-Scale Knowledge

**Linxi Fan**[1], **Guanzhi Wang**[2*], **Yunfan Jiang**[3*], **Ajay Mandlekar**[1], **Yuncong Yang**[4],
**Haoyi Zhu**[5], **Andrew Tang**[4], **De-An Huang**[1], **Yuke Zhu**[1 6†], **Anima Anandkumar**[1 2†]
[1]NVIDIA, [2]Caltech, [3]Stanford, [4]Columbia, [5]SJTU, [6]UT Austin
*Equal contribution   †Equal advising
https://minedojo.org

## Abstract

Autonomous agents have made great strides in specialist domains like Atari games
and Go. However, they typically learn *tabula rasa* in isolated environments with
limited and manually conceived objectives, thus failing to generalize across a wide
spectrum of tasks and capabilities. Inspired by how humans continually learn
and adapt in the open world, we advocate a trinity of ingredients for building
generalist agents: 1) an environment that supports a multitude of tasks and goals,
2) a large-scale database of multimodal knowledge, and 3) a flexible and scalable
agent architecture. We introduce MINEDOJO, a new framework built on the
popular *Minecraft* game that features a simulation suite with thousands of diverse
open-ended tasks and an internet-scale knowledge base with Minecraft videos,
tutorials, wiki pages, and forum discussions. Using MINEDOJO's data, we propose
a novel agent learning algorithm that leverages large pre-trained video-language
models as a learned reward function. Our agent is able to solve a variety of open-
ended tasks specified in free-form language without any manually designed dense
shaping reward. We open-source the simulation suite, knowledge bases, algorithm
implementation, and pretrained models (https://minedojo.org) to promote
research towards the goal of generally capable embodied agents.

## 1   Introduction

Developing autonomous embodied agents that can attain human-level performance across a wide
spectrum of tasks has been a long-standing goal for AI research. There has been impressive progress
towards this goal, most notably in games [62, 66, 96] and robotics [53, 74, 110, 101, 80]. These
embodied agents are typically trained *tabula rasa* in isolated worlds with limited complexity and
diversity. Although highly performant, they are specialist models that do not generalize beyond a
narrow set of tasks. In contrast, humans inhabit an infinitely rich reality, continuously learn from and
adapt to a wide variety of open-ended tasks, and are able to leverage large amount of prior knowledge
from their own experiences as well as others.

We argue that **three main pillars** are necessary for generalist embodied agents to emerge. First, the
environment in which the agent acts needs to **enable an unlimited variety of open-ended goals**
[88, 55, 92, 89]. Natural evolution is able to nurture an ever-expanding tree of diverse life forms
thanks to the infinitely varied ecological settings that the Earth supports [89, 98]. This process has
not stagnated for billions of years. In contrast, today's agent training algorithms cease to make new
progress after convergence in narrow environments [62, 110]. Second, a **large-scale database of
prior knowledge** is necessary to facilitate learning in open-ended settings. Just as humans frequently
learn from the internet, agents should also be able to harvest practical knowledge encoded in large
amounts of video demos [31, 59], multimedia tutorials [61], and forum discussions [97, 51, 41]. In a

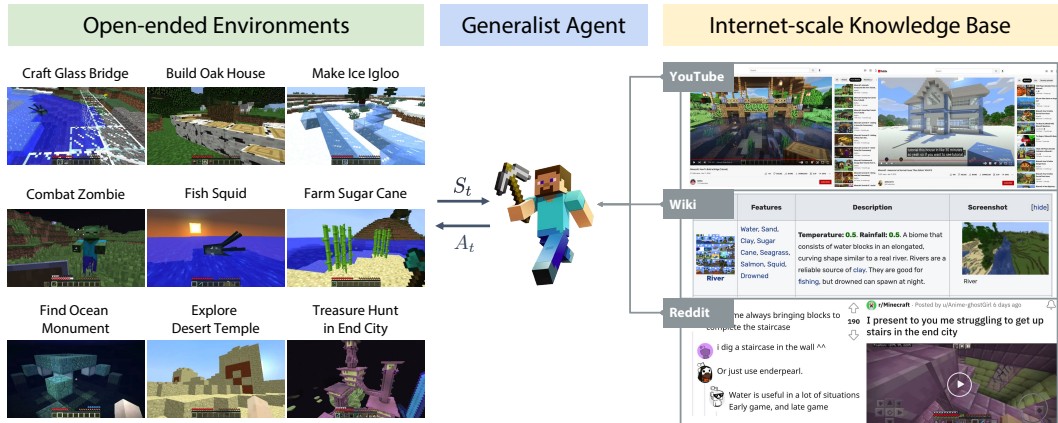

Figure 1: MINEDOJO is a novel framework for developing open-ended, generally capable agents that can learn and adapt continually to new goals. MINEDOJO features a benchmarking suite with *thousands* **of diverse open-ended tasks** specified in natural language prompts, and also provides an **internet-scale, multimodal knowledge base** of YouTube videos, Wiki pages, and Reddit posts. The database captures the collective experience and wisdom of millions of Minecraft gamers for an AI agent to learn from. Best viewed zoomed in.

complex world, it would be extremely inefficient for an agent to learn everything from scratch through trial and error. Third, the **agent's architecture** needs to be flexible enough to pursue any task in open-ended environments, and scalable enough to convert large-scale knowledge sources into actionable insights [16, 72]. This motivates the design of an agent that has a unified observation/action space, conditions on natural language task prompts, and adopts the Transformer pre-training paradigm [23, 68, 13] to internalize knowledge effectively.

In light of these three pillars, we introduce MINEDOJO, a new framework to help the community develop open-ended, generally-capable agents. It is built on the popular Minecraft game, where a player explores a procedurally generated 3D world with diverse types of terrains to roam, materials to mine, tools to craft, structures to build, and wonders to discover. Unlike most other games [62, 66, 96], Minecraft defines no specific reward to maximize and no fixed storyline to follow, making it well suited for developing open-ended environments for embodied AI research. We make the following three major contributions:

**1. Simulation platform with thousands of diverse open-ended tasks.** MINEDOJO provides convenient APIs on top of Minecraft that standardizes task specification, world settings, and agent's observation/action spaces. We introduce a benchmark suite that consists of thousands of natural language-prompted tasks, making it *two orders of magnitude* larger than prior Minecraft benchmarks like the MineRL Challenge [36, 49]. The suite includes long-horizon, open-ended tasks that cannot be easily evaluated through automated procedures, such as "*build an epic modern house with two floors and a swimming pool*". Inspired by the Inception score [73] and FID score [42] that are commonly used to assess AI-generated image quality, we introduce a novel agent evaluation protocol using a large video-language model pre-trained on Minecraft YouTube videos. This complements human scoring [78] that is precise but more expensive. Our learned evaluation metric has good agreement with human judgment in a subset of the full task suite considered in the experiments.

**2. Internet-scale multimodal Minecraft knowledge base.** Minecraft has more than 100 million active players [100], who have collectively generated an enormous wealth of data. They record tutorial videos, stream live play sessions, compile recipes, and discuss tips and tricks on forums. MINEDOJO features a massive collection of 730K+ YouTube videos with time-aligned transcripts, 6K+ free-form Wiki pages, and 340K+ Reddit posts with multimedia contents (Fig. 3). We hope that this enormous knowledge base can help the agent acquire diverse skills, develop complex strategies, discover interesting objectives, and learn actionable representations automatically.

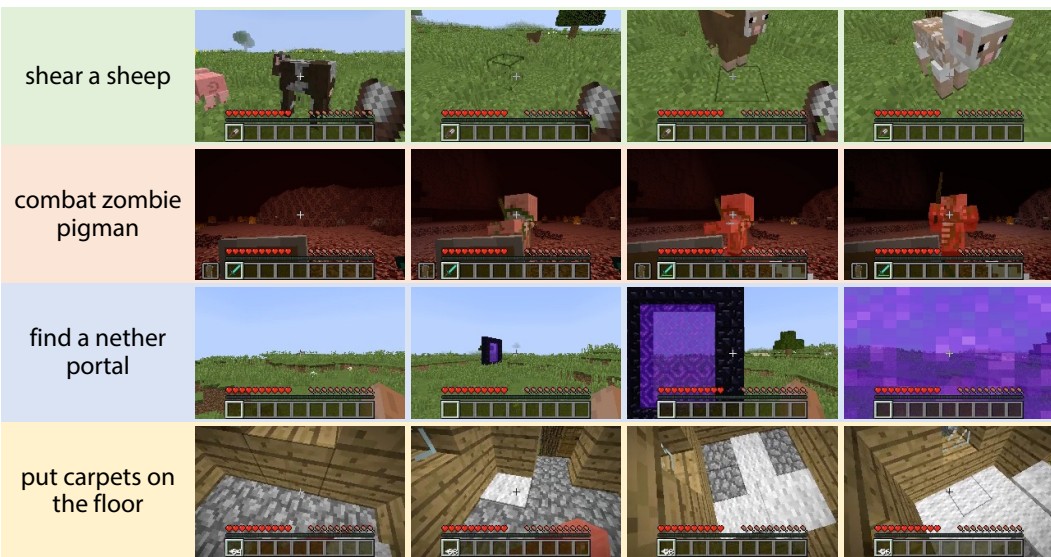

Figure 2: Visualization of our agent's learned behaviors on four selected tasks. Leftmost texts are the task prompts used in training. Best viewed on a color display.

**3. Novel algorithm for embodied agents with large-scale pre-training.** We develop a new learning algorithm for embodied agents that makes use of the internet-scale domain knowledge we have collected from the web. Using the massive volume of YouTube videos from MINEDOJO, we train a video-text contrastive model in the spirit of CLIP [69], which associates natural language subtitles with their time-aligned video segments. We demonstrate that this learned correlation score can be used effectively as an *open-vocabulary, massively multi-task reward function* for RL training. Our agent solves the majority of 12 tasks in our experiment using the learned reward model (Fig. 2). It achieves competitive performance to agents trained with meticulously engineered dense-shaping rewards, and in some cases outperforms them, with up to 73% improvement in success rates. For open-ended tasks that do not have a simple success criterion, our agents also perform well without any special modifications.

In summary, this paper proposes an open-ended task suite, internet-scale domain knowledge, and agent learning with recent advances on large pre-trained models [11]. We have open-sourced MINEDOJO's simulator, knowledge bases, algorithm implementations, pretrained model checkpoints, and task curation tools at `https://minedojo.org/`. We hope that MINEDOJO will serve as an effective starter framework for the community to develop new algorithms and advance towards generally capable embodied agent.

## 2 MINEDOJO Simulator & Benchmark Suite

MINEDOJO offers a set of simulator APIs help researchers develop generally capable, open-ended agents in Minecraft. It builds upon the open-source MineRL codebase [36] and makes the following upgrades: 1) We provide **unified observation and action spaces** across all tasks, facilitating the development of multi-task and continually learning agents that can constantly adapt to new scenarios and novel tasks. This deviates from the MineRL Challenge design that tailors observation and action spaces to individual tasks; 2) Our simulation unlocks all three types of worlds in Minecraft, including the *Overworld*, the *Nether*, and the *End*, which **substantially expands the possible task space**, while MineRL only supports the Overworld natively; and 3) We provide convenient APIs to configure initial conditions and world settings to standardize our tasks.

With this MINEDOJO simulator, we define thousands of benchmarking tasks, which are divided into two categories: 1) *Programmatic tasks* that can be automatically assessed based on the ground-truth simulator states; and 2) *Creative tasks* that do not have well-defined or easily-automated success criteria, which motivates our novel evaluation protocol using a learned model (Sec. 4). To scale up

the number of Creative tasks, we mine ideas from YouTube tutorials and use OpenAI's GPT-3 [13] service to generate substantially more task definitions. Compared to Creative tasks, Programmatic tasks are simpler to get started, but tend to have restricted scope, limited language variations, and less open-endedness in general.

## 2.1 Task Suite I: Programmatic Tasks

We formalize each programmatic task as a 5-tuple: $T = (G, \mathcal{G}, \mathcal{I}, f_{\mathcal{S}}, f_{\mathcal{R}})$. $G$ is an English description of the task goal, such as "*find material and craft a gold pickaxe*". $\mathcal{G}$ is a natural language guidance that provides helpful hints, recipes, or advice to the agent. We leverage OpenAI's GPT-3-davinci API to automatically generate detailed guidance for a subset of the tasks. For the example goal "*bring a pig into Nether*", GPT-3 returns: 1) Find a pig in the overworld; 2) Right-click on the pig with a lead; 3) Right-click on the Nether Portal with the lead and pig selected; 4) The pig will be pulled through the portal! $\mathcal{I}$ is the initial conditions of the agent and the world, such as the initial inventory, spawn terrain, and weather. $f_{\mathcal{S}}: s_t \rightarrow \{0, 1\}$ is the success criterion, a deterministic function that maps the current world state $s_t$ to a Boolean success label. $f_{\mathcal{R}}: s_t \rightarrow \mathbb{R}$ is an optional dense reward function. We only provide $f_{\mathcal{R}}$ for a small subset of the tasks in MINEDOJO due to the high costs of meticulously crafting dense rewards. For our current agent implementation (Sec. 4.1), we do not use detailed guidance. Inspired by concurrent works SayCan [3] and Socratic Models [107], one potential idea is to feed each step in the guidance to our learned reward model sequentially so that it becomes a stagewise reward function for a complex multi-stage task.

MINEDOJO provides 4 categories of programmatic tasks with 1,581 template-generated natural language goals to evaluate the agent's different capabilities systematically and comprehensively, including: 1) **Survival**: surviving for a designated number of days, 2) **Harvest**: finding, obtaining, cultivating, or manufacturing hundreds of materials and objects, 3) **Tech Tree**: crafting and using a hierarchy of tools, and 4) **Combat**: fighting various monsters and creatures that require fast reflex and martial skills. Each template has a number of variations based on the terrain, initial inventory, quantity, etc., which form a flexible spectrum of difficulty. In comparison, the NeurIPS MineRL Diamond challenge [36] is a subset of our programmatic task suite, defined by the task goal "*obtain 1 diamond*" in MINEDOJO.

## 2.2 Task Suite II: Creative Tasks

We define each creative task as a 3-tuple, $T = (G, \mathcal{G}, \mathcal{I})$, which differs from programmatic tasks due to the lack of straightforward success criteria. Inspired by model-based metrics like the Inception score [73] and FID score [42] for image generation, we design a novel task evaluation metric based on a pre-trained contrastive video-language model (Sec. 4.1). In the experiments, we find that the learned metric exhibits a high level of agreement with human evaluations (see Table 2).

We brainstormed and authored 216 Creative tasks, such as "*build a haunted house with zombie inside*" and "*race by riding a pig*". Nonetheless, such a manual approach is not scalable. Therefore, we develop two systematic approaches to extend the total number of task definitions to 1,560. This makes our Creative tasks *3 orders of magnitude* larger than Minecraft BASALT challenge [78], which has 4 Creative tasks.

**Approach 1. Task Mining from YouTube Tutorial Videos.** We identify our YouTube dataset as a rich source of tasks, as many human players demonstrate and narrate creative missions in the tutorial playlists. To collect high-quality tasks and accompanying videos, we design a 3-stage pipeline that makes it easy to find and annotate interesting tasks (see supplementary for details). Through this pipeline, we extract 1,042 task ideas from the common wisdom of a huge number of veteran Minecraft gamers, such as "*make an automated mining machine*" and "*grow cactus up to the sky*".

**Approach 2. Task Creation by GPT-3.** We leverage GPT-3's few-shot capability to generate new task ideas by seeding it with the tasks we manually authored or mined from YouTube. The prompt template is: Here are some example creative tasks in Minecraft: {a few examples}. Let's brainstorm more detailed while reasonable creative tasks in Minecraft. GPT-3 contributes 302 creative tasks after de-duplication, and demonstrates a surprisingly proficient understanding of Minecraft terminology.

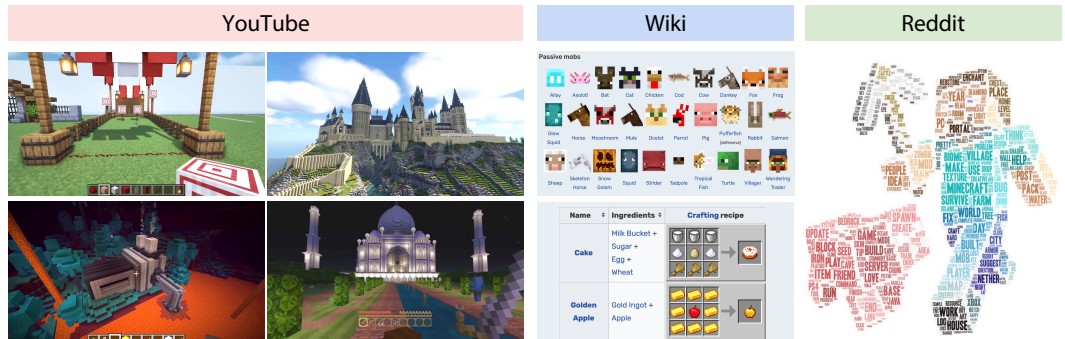

Figure 3: MINEDOJO's internet-scale, multimodal knowledge base. **Left, YouTube videos:** Minecraft gamers showcase the impressive feats they are able to achieve. Clockwise order: an archery range, Hogwarts castle, Taj Mahal, a Nether homebase. **Middle, Wiki:** Wiki pages contain multimodal knowledge in structured layouts, such as comprehensive catalogs of creatures and recipes for crafting. **Right, Reddit:** We create a word cloud from Reddit posts and comment threads. Gamers ask questions, share achievements, and discuss strategies extensively. Best viewed zoomed in.

## 2.3 Collection of Starter Tasks

We curate a set of 64 core tasks for future researchers to get started more easily. If their agent works well on these tasks, they can more confidently scale to the full benchmark. 1) **32 programmatic tasks**: 16 "standard" and 16 "difficult", spanning all 4 categories (survival, harvesting, combat, and tech tree). We rely on our Minecraft knowledge to decide the difficulty level. "Standard" tasks require fewer steps and lower resource dependencies to complete; and 2) **32 creative tasks**: 16 "standard" and 16 "difficult". Similarly, tasks labeled with "standard" are typically short-horizon tasks. We recommend that researchers run 100 evaluation episodes for each task and report the percentage success rate. The programmatic tasks have ground-truth success, while the creative tasks need our novel evaluation protocol (Sec. 5).

## 3 Internet-scale Knowledge Base

Two commonly used approaches [85, 96, 66, 27] to train embodied agents include training agents from scratch using RL with well-tuned reward functions for each task, or using a large amount of human-demonstrations to bootstrap agent learning. However, crafting well-tuned reward functions is challenging or infeasible for our task suite (Sec. 2.2), and employing expert gamers to provide large amounts of demonstration data would also be costly and infeasible [96].

Instead, we turn to the open web as an ever-growing, virtually unlimited source of learning material for embodied agents. The internet provides a vast amount of domain knowledge about Minecraft, which we harvest by extensive web scraping and filtering. We collect 33 years worth of YouTube videos, 6K+ Wiki pages, and millions of Reddit comment threads. Instead of hiring a handful of human demonstrators, we capture the collective wisdom of millions of Minecraft gamers around the world. Furthermore, language is a key and pervasive component of our database that takes the form of YouTube transcripts, textual descriptions in Wiki, and Reddit discussions. Language facilitates open-vocabulary understanding, provides grounding for image and video modalities, and unlocks the power of large language models [23, 82, 13] for embodied agents. To ensure socially responsible model development, we take special measures to filter out low-quality and toxic contents [11, 39] from our databases, detailed in the supplementary.

**YouTube Videos and Transcripts.** Minecraft is among the most streamed games on YouTube [30]. Human players have demonstrated a stunning range of creative activities and sophisticated missions that take hours to complete (examples in Fig. 3). We collect 730K+ narrated Minecraft videos, which add up to 33 years of duration and 2.2B words in English transcripts. In comparison, HowTo100M [59] is a large-scale human instructional video dataset that includes 15 years of experience in total – about half of our volume. The time-aligned transcripts enable the agent to ground free-form natural lan-

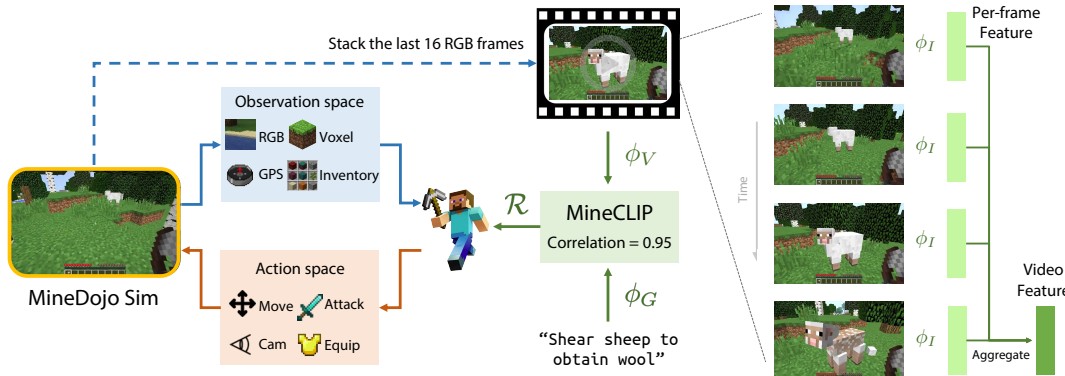

Figure 4: Algorithm design. MINECLIP is a contrastive video-language model pre-trained on MINEDOJO's massive Youtube database. It computes the correlation between an open-vocabulary language goal string and a 16-frame video snippet. The correlation score can be used as a learned dense reward function to train a strong multi-task RL agent.

guage in video pixels and learn the semantics of diverse activities without laborious human labeling. We operationalize this insight in our pre-trained video-language model (Sec. 4.1).

**Minecraft Wiki.** The Wiki pages cover almost every aspect of the game mechanics, and supply a rich source of unstructured knowledge in multimodal tables, recipes, illustrations, and step-by-step tutorials. We use Selenium [77] to scrape 6,735 pages that interleave text, images, tables, and diagrams. The pages are highly unstructured and do not share any common schema, as the Wiki is meant for human consumption rather than AI training. To preserve the layout information, we additionally save the screenshots of entire pages and extract 2.2M bounding boxes of the visual elements (visualization in supplementary). We do not use Wiki data in our current experiments. Since the Wiki contains detailed recipes for all crafted objects, they could be provided as input or training data for hierarchical planning methods and policy sketches [7]. Another promising future direction is to apply document understanding models such as LayoutLM [105, 104] and DocFormer [8] to learn actionable knowledge from these unstructured Wiki data.

**Reddit.** We scrape 340K+ posts along with 6.6M comments under the "r/Minecraft" subreddit. These posts ask questions on how to solve certain tasks, showcase cool architectures and achievements in image/video snippets, and discuss general tips and tricks for players of all expertise levels. We do not use Reddit data to train our current agent. A potential idea is to finetune large language models [23, 68] on our Reddit corpus to generate instructions and execution plans that are better grounded in the Minecraft domain. Concurrent works [3, 43, 107] have explored similar ideas and showed excellent results on robot learning, which is encouraging for more future research in MINEDOJO.

## 4    Agent Learning with Large-scale Pre-training

One of the grand challenges of embodied AI is to build a single agent that can complete a wide range of open-world tasks. The MINEDOJO framework aims to facilitate new techniques towards this goal by providing an open-ended task suite (Sec. 2) and large-scale internet knowledge base (Sec. 3). Here we take an initial step towards this goal by developing a proof of concept that demonstrates how a single language-prompted agent can be trained in MINEDOJO to complete several complex Minecraft tasks. To this end, we propose a novel agent learning algorithm that takes advantage of the massive YouTube data offered by MINEDOJO. We note that this is only one of the numerous possible ways to use MINEDOJO's internet database — the Wiki and Reddit corpus also hold great potential to drive new algorithm discoveries for the community in future works.

In this paper, we consider a multi-task reinforcement learning (RL) setting, where an agent is tasked with completing a collection of MINEDOJO tasks specified by language instructions (Sec. 2). Solving these tasks often requires the agent to interact with the Minecraft world in a prolonged fashion. Agents developed in popular RL benchmarks [91, 110] often rely on meticulously crafted dense and

Table 1: Our novel MINECLIP reward model is able to achieve competitive performance with manually written dense reward function for Programmatic tasks, and significantly outperforms the CLIP$_{OpenAI}$ method across all Creative tasks. Entries represent percentage success rates averaged over 3 seeds, each tested for 200 episodes. Success conditions are precise in Programmatic tasks, but estimated by MineCLIP for Creative tasks.

| Group | Tasks | **Ours** (Attn) | **Ours** (Avg) | Manual Reward | Sparse-only | CLIP$_{OpenAI}$ |
|---|---|---|---|---|---|---|
| | Milk Cow | **64.5 ± 37.1** | 6.5 ± 3.5 | 62.8 ± 40.1 | 0.0 ± 0.0 | 0.0 ± 0.0 |
| | Hunt Cow | **83.5 ± 7.1** | 0.0 ± 0.0 | 48.3 ± 35.9 | 0.3 ± 0.4 | 0.0 ± 0.0 |
| | Shear Sheep | 12.1 ± 9.1 | 0.6 ± 0.2 | **52.3 ± 33.2** | 0.0 ± 0.0 | 0.0 ± 0.0 |
| | Hunt Sheep | 8.1 ± 4.1 | 0.0 ± 0.0 | **41.9 ± 33.0** | 0.3 ± 0.4 | 0.0 ± 0.0 |
| | Combat Spider | 80.5 ± 13.0 | 60.1 ± 42.5 | **87.5 ± 4.6** | 47.8 ± 33.8 | 0.0 ± 0.0 |
| | Combat Zombie | 47.3 ± 10.6 | **72.3 ± 6.4** | 49.8 ± 26.9 | 8.8 ± 12.4 | 0.0 ± 0.0 |
| | Combat Pigman | 1.6 ± 2.3 | 0.0 ± 0.0 | **13.6 ± 9.8** | 0.0 ± 0.0 | 0.0 ± 0.0 |
| | Combat Enderman | 0.0 ± 0.0 | 0.0 ± 0.0 | 0.3 ± 0.2 | 0.0 ± 0.0 | 0.0 ± 0.0 |
| | Find Nether Portal | 37.4 ± 40.8 | **89.8 ± 5.7** | N/A | N/A | 26.3 ± 32.6 |
| | Find Ocean | 33.4 ± 45.6 | **54.3 ± 40.7** | N/A | N/A | 9.9 ± 14.1 |
| | Dig Hole | **91.6 ± 5.9** | 88.1 ± 13.3 | N/A | N/A | 0.0 ± 0.0 |
| | Lay Carpet | 97.6 ± 1.9 | **98.8 ± 1.0** | N/A | N/A | 0.0 ± 0.0 |

task-specific reward functions to guide random explorations. However, these rewards are hard or even infeasible to define for our diverse and open-ended tasks in MINEDOJO. To address this challenge, our key insight is to learn **a dense, language-conditioned reward function from in-the-wild YouTube videos and their transcripts**. Therefore, we introduce **MINECLIP**, a contrastive video-language model that learns to correlate video snippets and natural language descriptions (Fig. 4). MINECLIP is multi-task by design, as it is trained on open-vocabulary and diverse English transcripts.

During RL training, MINECLIP provides a high-quality reward signal *without* any domain adaptation techniques, despite the domain gap between noisy YouTube videos and clean simulator-rendered frames. MINECLIP eliminates the need to manually engineer reward functions for each and every MINEDOJO task. For Creative tasks that lack a simple success criterion (Sec. 2.2), MINECLIP also serves the dual purpose of an **automatic evaluation metric** that agrees well with human judgement on a subset of tasks we investigate (Sec. 4.2, Table 2). Because the learned reward model incurs a non-trivial computational overhead, we introduce several techniques to significantly improve RL training efficiency, making MINECLIP a practical module for open-ended agent learning in Minecraft (Sec. 4.2).

## 4.1 Pre-Training MINECLIP on Large-scale Videos

Formally, the learned reward function can be defined as $\Phi_{\mathcal{R}} : (G, V) \to \mathbb{R}$ that maps a language goal $G$ and a video snippet $V$ to a scalar reward. An ideal $\Phi_{\mathcal{R}}$ should return a high reward if the behavior depicted in the video faithfully follows the language description, and a low reward otherwise. This can be achieved by optimizing the InfoNCE objective [95, 40, 17], which learns to correlate positive video and text pairs [90, 5, 60, 4, 103].

Similar to the image-text CLIP model [69], MINECLIP is composed of a separate text encoder $\phi_G$ that embeds a language goal and a video encoder $\phi_V$ that embeds a moving window of 16 consecutive frames with $160 \times 256$ resolution (Fig. 4). Our neural architecture has a similar design as CLIP4Clip [58], where $\phi_G$ reuses OpenAI CLIP's pretrained text encoder, and $\phi_V$ is factorized into a frame-wise image encoder $\phi_I$ and a temporal aggregator $\phi_a$ that summarizes the sequence of 16 image features into a single video embedding. Unlike CLIP4Clip, we insert two extra layers of residual CLIP Adapter [29] after the aggregator $\phi_a$ to produce a better video feature, and finetune *only* the last two layers of the pretrained $\phi_I$ and $\phi_G$.

From the MINEDOJO YouTube database, we follow the procedure in VideoCLIP [103] to sample 640K pairs of 16-second video snippets and time-aligned English transcripts, after applying a keyword filter. We train two MINECLIP variants with different types of aggregator $\phi_a$: (1) MINECLIP[avg] does simple average pooling, which is fast but agnostic to the temporal ordering; (2) MINECLIP[attn] encodes the sequence by two transformer layers, which is relatively slower but captures more temporal

Table 2: MINECLIP agrees well with the ground-truth human judgment on the Creative tasks we consider. Numbers are F1 scores between MINECLIP's binary classification of tasks success and human labels (scaled to the percentage for better readability).

| Tasks | Find Nether Portal | Find Ocean | Dig Hole | Lay Carpet |
|---|---|---|---|---|
| Ours (Attn) | 98.7 | **100.0** | 99.4 | 97.4 |
| Ours (Avg) | **100.0** | **100.0** | **100.0** | **98.4** |
| CLIP$_{\text{OpenAI}}$ | 48.7 | 98.4 | 80.6 | 54.1 |

information, and thus produces a better reward signal in general. Details of data preprocessing, architecture, and hyperparameters are in the supplementary.

### 4.2 RL with MINECLIP Reward

We train a language-conditioned policy network that takes as input raw pixels and predicts discrete control. The policy is trained with PPO [76] on the MINECLIP rewards. In each episode, the agent is prompted with a language goal and takes a sequence of actions to fulfill this goal. When calculating the MINECLIP rewards, we concatenate the agent's latest 16 egocentric RGB frames in a temporal window to form a video snippet. MINECLIP handles all task prompts *zero-shot* without any further finetuning. In our experiments (Sec. 5), we show that MINECLIP provides effective dense rewards out of the box, despite the domain shift between in-the-wild YouTube frames and simulator frames. Besides regular video data augmentation, we do not employ any special domain adaptation methods during pre-training. Our finding is consistent with CLIP's strong zero-shot performances on robustness benchmarks in object recognition [69].

Compared to hard-coded reward functions in popular benchmarks [110, 91, 26], the MINECLIP model has 150M parameters and is thus much more expensive to query. We make several design choices to greatly accelerate RL training with MINECLIP in the loop: 1) The language goal $G$ is fixed for a specific task, so the **text features** $\phi_G$ **can be precomputed** to avoid invoking the text encoder repeatedly; 2) Our agent's **RGB encoder reuses the pre-trained weights of** $\phi_I$ from MINECLIP. We do not finetune $\phi_I$ during RL training, which saves computation and endows the agent with good visual representations from the beginning; 3) MINECLIP's video encoder $\phi_V$ is factorized into an image encoder $\phi_I$ and a light-weight aggregator $\phi_a$. This design choice enables **efficient image feature caching**. Consider two overlapping video sequences of 8 frames, `V[0:8]` and `V[1:9]`. We can cache the image features of the 7 overlapping frames `V[1]` to `V[7]` to maximize compute savings. If $\phi_V$ is a monolithic model like S3D [102] in VideoCLIP [103], then the video features from every sliding window must be recomputed, which would incur a much higher cost per time step; and 4) We leverage **Self-Imitation Learning** [65] to store the trajectories with high MINECLIP reward values in a buffer, and alternate between PPO and self-imitation gradient steps. It further improves sample efficiency as shown in the supplementary materials.

## 5 Experiments

We evaluate our agent-learning approach (Section 4) on 8 Programmatic tasks and 4 Creative tasks from the MINEDOJO benchmarking suite. We select these 12 tasks due to the diversity of skills required to solve them (e.g., harvesting, combat, building, navigation) and domain-specific entities (e.g., animals, resources, monsters, terrains, and structures). We split the tasks into 3 groups and train one multi-task agent for each group: `Animal-Zoo` (4 Programmatic tasks on hunting or harvesting resource from animals), `Mob-Combat` (Programmatic, fight 4 types of hostile monsters), and `Creative` (4 tasks).

In the experiments, we empirically check the quality of MINECLIP against manually written reward functions, and quantify how different variants of our learned model affect the RL performance. Table 1 presents our main results. Policy networks of all methods share the same architecture and are trained by PPO + Self-Imitation (Sec. 4.2). We compare the following methods: 1) **Ours (Attn)**: our agent trained with the MINECLIP[attn] reward model. For Programmatic tasks, we also add the final success condition as a binary reward. For Creative tasks, MINECLIP is the only source of reward; 2) **Ours (Avg)**: the average-pooling variant; 3) **Manual Reward**: hand-engineered dense reward using

Table 3: MINECLIP agents have stronger zero-shot visual generalization ability to unseen terrains, weathers, and lighting. Numbers outside parentheses are percentage success rates averaged over 3 seeds (each tested for 200 episodes), while those inside parentheses are relative performance changes.

| | Tasks | **Ours** (Attn), train | **Ours** (Attn), unseen test | $\text{CLIP}_{\text{OpenAI}}$, train | $\text{CLIP}_{\text{OpenAI}}$, unseen test |
|---|---|---|---|---|---|
| | Milk Cow | $64.5 \pm 37.1$ | $\mathbf{64.8 \pm 31.3}(+\ 0.8\%)$ | $90.0 \pm 0.4$ | $29.2 \pm 3.7\ (-67.6\%)$ |
| | Hunt Cow | $83.5 \pm 7.1$ | $\mathbf{55.9 \pm 7.2}\ (-32.9\%)$ | $72.7 \pm 3.5$ | $16.7 \pm 1.6\ (-77.0\%)$ |
| | Combat Spider | $80.5 \pm 13.0$ | $\mathbf{62.1 \pm 29.7}(-22.9\%)$ | $79.5 \pm 2.5$ | $54.2 \pm 9.6\ (-31.8\%)$ |
| | Combat Zombie | $47.3 \pm 10.6$ | $\mathbf{39.9 \pm 25.3}(-15.4\%)$ | $50.2 \pm 7.5$ | $30.8 \pm 14.4(-38.6\%)$ |

ground-truth simulator states; 4) **Sparse-only**: the final binary success as a single sparse reward. Note that neither sparse-only nor manual reward is available for Creative tasks; and 5) **CLIP$_{\text{OpenAI}}$**: pre-trained OpenAI CLIP model that has **not** been finetuned on any MINEDOJO videos. All RL training details are presented in the supplementary. Fig. 2 visualizes the learned agent behavior in 4 of the considered tasks.

**MINECLIP is competitive with manual reward.** For Programmatic tasks (first 8 rows), RL agents guided by MINECLIP achieve competitive performance as those trained by manual reward. In three of the tasks, they even *outperform* the hand-engineered reward functions, which rely on privileged simulator states unavailable to MINECLIP. For a more statistically sound analysis, we conduct the Paired Student's *t*-test to compare the mean success rate of each task (pairing column 3 "Ours (Attn)" and column 5 "Manual Reward" in Table 1). The test yields p-value $0.3991 \gg 0.05$, which indicates that the difference between our method and manual reward is not considered statistically significant, and therefore they are comparable with each other. Because all tasks require nontrivial exploration, our approach also dominates the Sparse-only baseline. Note that the original OpenAI CLIP model fails to achieve any success. We hypothesize that the creatures in Minecraft look dramatically different from their real-world counterparts, which causes CLIP to produce *misleading* signals worse than no shaping reward at all. It implies the importance of finetuning on MINEDOJO's YouTube data.

**MINECLIP provides automated evaluation.** For Creative tasks (last 4 rows), there are no programmatic success criteria available. We convert MINECLIP into a binary success classifier by thresholding the reward value it outputs for an episode. To test the quality of MINECLIP as an automatic evaluation metric, we ask human judges to curate a dataset of 100 successful and 100 failed trajectories for each task. We then run both MINECLIP variants and CLIP$_{\text{OpenAI}}$ on the dataset and report the binary F1 score of their judgement against human ground-truth in Table 2. The results demonstrate that both MINECLIP[attn] and MINECLIP[avg] attain a very high degree of agreement with human evaluation results on this subset of the Creative task suite that we investigate. CLIP$_{\text{OpenAI}}$ baseline also achieves nontrivial agreement on Find Ocean and Dig Hole tasks, likely because real-world oceans and holes have similar texture. We use the `attn` variant as an automated success criterion to score the 4 Creative task results in Table 1. Our proposed method consistently learns better than CLIP$_{\text{OpenAI}}$-guided agents. It shows that MINECLIP is an effective approach to solving open-ended tasks when no straightforward reward signal is available. We provide further analysis beyond these 4 tasks in the supplementary.

**MINECLIP shows good zero-shot generalization to significant visual distribution shift.** We evaluate the learned policy without finetuning on a combination of unseen weather, lighting conditions, and terrains — 27 scenarios in total. For the baseline, we train agents with the original CLIP$_{\text{OpenAI}}$ image encoder (not trained on our YouTube videos) by imitation learning. The robustness against visual shift can be quantitatively measured by the relative performance degradation on novel test scenarios for each task. Table 3 shows that while all methods incur performance drops, agents with MINECLIP encoder is more robust to visual changes than the baseline across all tasks. Pre-training on diverse in-the-wild YouTube videos is important to boosting zero-shot visual generalization capability, a finding consistent with literature [69, 64].

## 6 Related work

**Open-ended Environments for Decision-making Agents.** There are many environments developed with the goal of open-ended agent learning. Prior works include maze-style worlds

[93, 98, 48], purely text-based game [54], grid worlds [18, 14], browser/GUI-based environments [81, 94], and indoor simulators for robotics [1, 80, 87, 26, 83, 74, 67]. Minecraft offers an exciting alternative for open-ended agent learning. It is a 3D visual world with procedurally generated landscapes and extremely flexible game mechanics that support an enormous variety of activities. Prior methods in open-ended agent learning [25, 44, 99, 50, 22] do not make use of external knowledge, but our approach leverages internet-scale database to learn open-vocabulary reward models, thanks to Minecraft's abundance of gameplay data online.

**Minecraft for AI Research.** The Malmo platform [47] is the first comprehensive release of a Gym-style agent API [12] for Minecraft. Based on Malmo, MineRL [36] provides a codebase and human play trajectories for the annual Diamond Challenge at NeurIPS [35, 37, 49]. MINEDOJO's simulator builds upon the pioneering work of MineRL, but greatly expands the API and benchmarking task suite. Other Minecraft benchmarks exist with different focuses. For example, CraftAssist [33] and IGLU [52] study agents with interactive dialogues. BASALT [78] applies human evaluation to 4 open-ended tasks. EvoCraft [34] is designed for structure building, and Crafter [38] optimizes for fast experimentation. Unlike prior works, MINEDOJO's core mission is to facilitate the development of generally capable embodied agents using internet-scale knowledge. We include a feature comparison table of different Minecraft platforms for AI research in the supplementary.

**Internet-scale Multimodal Knowledge Bases.** Big dataset such as Common Crawl [20], the Pile [28], LAION [75], YouTube-8M [2] and HowTo100M [59] have been fueling the success of large pre-trained language models [23, 68, 13] and multimodal models [90, 5, 60, 109, 6, 4, 103]. While generally useful for learning representations, these datasets are not specifically targeted at embodied agents. To provide agent-centric training data, RoboNet [21] collects video frames from 7 robot platforms, and Ego4D [32] recruits volunteers to record egocentric videos of household activities. In comparison, MINEDOJO's knowledge base is constructed without human curation efforts, much larger in volume, more diverse in data modalities, and comprehensively covers all aspects of the Minecraft environment.

**Embodied Agents with Large-scale Pre-training.** Inspired by the success in NLP, embodied agent research [24, 10, 70, 19] has seen a surge in adoption of the large-scale pre-training paradigm. The recent advances can be roughly divided into 4 categories. 1) **Novel agent architecture**: Decision Transformer [16, 45, 108] applies self-attention to sequential decision making. GATO [71] and Unified-IO [57] learn a single model to accommodate different control interfaces. VIMA [46] unifies a wide range of robot manipulation tasks with multimodal prompting. 2) **Pre-training for better representations**: R3M [64] trains a general-purpose visual encoder for robot perception on Ego4D videos [32]. CLIPort [84] leverages the pre-trained CLIP model [69] to enable free-form language instructions for robot manipulation. 3) **Pre-training for better policies**: AlphaStar [96] achieves champion-level performance on StarCraft by imitating from numerous human demos. SayCan [3] leverages large language models (LMs) to ground value functions in the physical world. [56] and [72] directly reuse pre-trained LMs as policy backbone. VPT [9] is a concurrent work that learns an inverse dynamics model from human contractors to pseudo-label YouTube videos for behavior cloning. VPT is complementary to our approach, and can be finetuned to solve language-conditioned open-ended tasks with our learned reward model. 4) **Data-driven reward functions**: Concept2Robot [79] and DVD [15] learn a binary classifier to score behaviors from in-the-wild videos [31]. LOReL [63] crowd-sources humans labels to train language-conditioned reward function for offline RL. AVID [86] and XIRL [106] extract reward signals via cycle consistency. MINEDOJO's task benchmark and internet knowledge base are generally useful for developing new algorithms in all the above categories. In Sec. 4, we propose an open-vocabulary, multi-task reward model using MINEDOJO YouTube videos.

## 7  Conclusion

In this work, we introduce the MINEDOJO framework for developing generally capable embodied agents. MINEDOJO features a benchmarking suite of thousands of Programmatic and Creative tasks, and an internet-scale multimodal knowledge base of videos, wiki, and forum discussions. As an example of the novel research possibilities enabled by MINEDOJO, we propose MINECLIP as an effective language-conditioned reward function trained with in-the-wild YouTube videos. MINECLIP achieves strong performance empirically and agrees well with human evaluation results, making it a good automatic metric for Creative tasks. We look forward to seeing how MINEDOJO empowers the community to make progress on the important challenge of open-ended agent learning.

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
