# OpenReview forum: "MineDojo: Building Open-Ended Embodied Agents with Internet-Scale Knowledge"
_NeurIPS.cc/2022/Track/Datasets_and_Benchmarks — NeurIPS 2022 Datasets and Benchmarks _

### Official Review · Reviewer_Z7oY · 2022-07-11
**MineDOJO Review**

**Rating:** 8
**Confidence:** 4

**Strengths:**

The large amount of diversity in the task space, the open-endedness of the underlying environment, the large amount of easy-to-use text, image and video data are all major strengths of the contribution. The paper is well-written (although many details are left to the appendix, which is understandable given the amount of information presented). Demonstrating an initial approach in MineCLIP is beneficial to enable future research, as there is an existing solid baseline.

**Weaknesses:**

There isn't much time spend on recommended evaluation protocols for future researchers. It would be beneficial to define (possibly several) fixed task selections, so that future researchers will have comparable results. This will encourage more robust research, and disincentivise cherry-picking tasks that future methods perform well on.

While there is a focus in the paper and dataset on open-ended learning, a key aspect of open-ended embodied AI is generalisation, and this doesn't seem to be discussed much or tackled by the benchmark as-is. I think Minecraft is a great domain to test generalisation due to the underlying procedural generation, and it would be great to leverage that to enable evaluating how general the learned policies are. For example, for simple tasks that aren't too dependent on the exact features of the terrain, randomising over procedurally generated environments (with fixed seeds) during training, and then evaluating on held-out environments (as in OpenAI Procgen for example) would enable this. A stronger form of task generalisation would tie in with the above recommendation: a subset of tasks could be chosen for training, and then models could be evaluated on their generalisation to new tasks zero-shot (or after a small amount of fine-tuning, if zero-shot performance isn't good).

I think if these recommendations and weaknesses are addressed (which seems feasible within the rebutal period) then I'd be happy to raise my score to an 8.


**Additional Feedback:**

How are tasks chosen for evaluation? Did you try multi-task training with more tasks?

How should we understand the reward numbers in Table 1? Are they normalised? If not, could they be? Could you present an average (normalised) reward across all programmatic tasks, so that we can validate whether your approach is "competitive" with the hand-designed reward functions in a statistical way. This could include using the tools from RLiable for more robust evaluation, given you train for multiple seeds over multiple tasks.

Could you describe Table 2 in it's caption, in addition to drawing conclusions from it?



**Clarity:**

The paper is clear and well-written, apart from the concerns raised in the correctness section above.

**Correctness:**

Generally the work is correct in its claims, given the experimental results. However, there are some misleading claims that I think aren't sufficiently cautious. For example, it's repeatedly claimed that the MineCLIP method is "competitive" with the hand-crafted reward functions, but there's not a statistical comparison to back up this fact. Secondly, in the intro (and in other places) it's claimed that "Our learned evaluation metric agrees with human judgement in > 97% of test cases and is consistently high-quality across different tasks", but this is only validated on 4 of the creative tasks, all of which seem on the relatively simple side. While it's not surprising that the MineCLIP approach works for these simple tasks (that can effectively checked in only a few frames), the more complex creative tasks such as building a haunted house or a mansion seem much harder to evaluate. It would be beneficial to investigate whether the MineCLIP reward function generalises well to these cases.

**Documentation:**

The documentation is thorough and complete.

**Ethics:**

The only ethical concern I can see is with the scraping of internet data from the minecraft wiki, reddit and youtube. Are the licenses of these data sources permissive enough to allow this scraping. Particularly in the case of YouTube, I imagine that many of the videos come from youtubers who make money from ads on those videos.

**Relation To Prior Work:**

Prior work is generally well-cited. However, I think it would be beneficial to cite work on open-ended algorithms, such as POET, PAIRED and other works in this space, and contrasting their approach to open-ended agents with the approach presented in this paper.

**Summary And Contributions:**

This paper proposes the MineDOJO benchmark, which comes with an accompanying dataset. The benchmark is a selection of around 3100 tasks in the Minecraft simulator, each of which is described a natural-language prompt. The tasks are split into two categories: programmatic tasks, which have a ground truth reward function based on the simulator state; and creative tasks, that don't have a ground truth reward function and are more open-ended. The accompanying dataset has a large selection of youtube videos about minecraft, with time-aligned text transcripts, as well as around 7000 freeform minecraft wiki pages and around 350k reddit posts from r/Minecraft. The work propose using the video-text data to learn a CLIP-style model that learns to match open-vocabulary natural text descriptions (from the transcripts) to 16-frame video segments. This trained model is then used as a reward function for reinforcement learning, and also as an evaluation mechanism, especially for the creative tasks where there is no alternative. They show that this evaluation mechanism is fairly well-correlated with human evaluations of task completion, and show that using this reward function combined with PPO with self imitation learning enables the learning of language-conditioned policies which get some reward, and that their method performs better than just using plain CLIP (from OpenAI) as the reward function.

---

> ### Author Response · Authors · 2022-08-18
> **Response (3/3)**
>
> > _*"in the intro (and in other places) it's claimed that "Our learned evaluation metric agrees with human judgement in > 97% of test cases and is consistently high-quality across different tasks", but this is only validated on 4 of the creative tasks, all of which seem on the relatively simple side. [...] the more complex creative tasks such as building a haunted house or a mansion seem much harder to evaluate. It would be beneficial to investigate whether the MineCLIP reward function generalises well to these cases."*_
>
> Thanks for the feedback! For this rebuttal, we have further annotated 50 YouTube video segments each for 5 more tasks that are much more semantically complex and run MineCLIP evaluation on these videos against a negative set. The agreement with human annotation is measured in binary classification F1 scores below (scaled by 100 for better readability).
>
> | Tasks | Build a farm | Build a fence | Build a house | Ride a minecart | Build a swimming pool |
> |:---:|:---:|:---:|:---:|:---:|:---:|
> | Ours (attn) | 78.7 | 91.4 | 63.7 | 95.9 | 85.0 |
> | Ours (avg) | 73.4 | 83.1 | 37.4 | 96.9 | 94.7 |
>
> We can see that MineCLIP generally has a positive agreement with human judgment, but it is not perfect. We note that the current MineCLIP is a proof-of-concept step in leveraging internet data for automated evaluation, and further scaling on more training data and parameters may lead to more improvements. Meanwhile, human judgment remains a useful and important alternative [9,10]. Following your advice, we have toned down the claim in our revised paper (Sec. 6) and added the above results to the supplementary (Sec. 9.3).
>
> * [9] Hierarchical Text-Conditional Image Generation with CLIP Latents. Ramesh et al. 2022.
> * [10] Photorealistic Text-to-Image Diffusion Models with Deep Language Understanding. Saharia et al. 2022.
>
> > _*"Prior work is generally well-cited. However, I think it would be beneficial to cite work on open-ended algorithms, such as POET, PAIRED and other works in this space, and contrasting their approach to open-ended agents with the approach presented in this paper."*_
>
> Thanks for the suggestion! POET has already been cited, but we have added PAIRED [11] to the Related Work section. These methods are prior SOTAs on open-ended agent learning. The main difference is that our approach leverages internet-scale video data to learn open-vocabulary reward functions, while prior works do not make use of external knowledge.
>
> * [11] Emergent Complexity and Zero-shot Transfer via Unsupervised Environment Design. Dennis et al. 2020.
>
> > _*"How are tasks chosen for evaluation? Did you try multi-task training with more tasks?"*_
>
> Great question. We selected these 12 tasks due to the diversity of skills required to solve them (e.g., harvesting, combat, building, navigation) and domain-specific entities (e.g., animals, resources, monsters, terrains, and structures). Mentioned in a previous point (new Table 6, Sec. 9.1, supplementary), we have trained a single agent to learn all 12 tasks and showed zero-shot and finetuning performance on novel held-out tasks.
>
> > _*"Could you describe Table 2 in it's caption, in addition to drawing conclusions from it?"*_
>
> Sorry for the confusion! In Table 2, the numbers are F1 scores between MineCLIP’s binary classification of tasks success and human labels. We scale it by 100 for better readability. We have revised the Table 2 caption to make this clear.

---

> ### Author Response · Authors · 2022-08-18
> **Response (2/3)**
>
> > _*"A stronger form of task generalisation would tie in with the above recommendation: a subset of tasks could be chosen for training, and then models could be evaluated on their generalisation to new tasks zero-shot (or after a small amount of fine-tuning, if zero-shot performance isn't good)."*_
>
> We appreciate the suggestion! Following your advice, we trained a single multitask agent on all 12 tasks (results reported in the newly added Table 6 in the supplementary, Sec. 9.1).
> We then evaluate this agent on two novel tasks: “harvest spider string” and “hunt pig”. The table is presented below (also as Table 7, supplementary). Results show that the agent struggles in the zero-shot setting because it has not interacted with pigs or spider strings during training, and thus does not know how to handle them effectively. However, the performance improves substantially by finetuning with the MineCLIP reward. Here the baseline methods are trained from scratch using RL with MineCLIP encoders and reward. Therefore, the only difference is whether the policy has been pre-trained on the 12 tasks or not. Given the same environment sampling budget, our method significantly outperforms the baseline. It suggests that the multitask agent has learned transferable knowledge on hunting and resource collection, which enables it to quickly adapt to previously unseen tasks.
>
> |  | Hunt Pig | Spider String |
> |:---:|:---:|:---:|
> | Ours (zero-shot) | 1.3 +- 0.6 | 1.6 +- 1.3 |
> | Ours (after RL finetune) | 46.0 +- 15.3 | 36.5 +- 16.9 |
> | Baseline (RL from scratch) | 0.0 +- 0.0 | 12.5 +- 12.7 |
>
> > _*"How should we understand the reward numbers in Table 1? Are they normalised? If not, could they be?"*_
>
> Sorry for the confusion! In Table 1, the numbers are percentage success rates averaged over 3 random training seeds, each tested for 200 episodes. Thus, their range is between 0% and 100% by definition. For programmatic tasks, the success rates are precise. For creative tasks, the success condition is determined by converting MineCLIP into a binary classifier (procedure explained in Sec 8.4, supplementary). We have revised the caption of Table 1 to make this clear.
>
> > _*"It's repeatedly claimed that the MineCLIP method is "competitive" with the hand-crafted reward functions, but there's not a statistical comparison to back up this fact."*_
>
> > _*"Could you present an average (normalised) reward across all programmatic tasks, so that we can validate whether your approach is "competitive" with the hand-designed reward functions in a statistical way."*_
>
> Thanks for the suggestion! We will be more cautious in our wording. To systematically compare MineCLIP with the hand-crafted reward function, we adopt the Paired Student’s t-test to compare the mean success rate of each task (pairing column 3 “MineCLIP[attn]” and column 5 “Manual Reward” in Table 1). The test yields p-value $0.3991 \gg  0.05$, which indicates that the difference between our method and manual reward is not considered statistically significant, and therefore they are comparable with each other.
> We have added this statistical analysis to Sec. 6 of the main paper.
>
> **<CONTINUED>**

---

> ### Author Response · Authors · 2022-08-18
> **Response (1/3)**
>
> Dear Reviewer,
>
> We truly appreciate your thorough review, positive feedback, and time and efforts taken to help us strengthen the paper even more! We are delighted that you find our paper clear and well-written, our benchmark useful, and our MineCLIP method a solid baseline. We have conducted more experiments and ablations that you suggested, revised the paper accordingly, and will address your concerns in turn below:
>
> > _*"There isn't much time spend on recommended evaluation protocols for future researchers. It would be beneficial to define (possibly several) fixed task selections, so that future researchers will have comparable results."*_
>
> This is a great suggestion! We followed your advice and curated a set of [64 tasks](https://github.com/MineDojo/MineDojo/blob/main/minedojo/tasks/description_files/tasks_suite.yaml) for future researchers to get started more easily. If their agent works well on these starter tasks, they can more confidently scale to the full benchmark. The composition is as follows:
>
> * 32 programmatic tasks: 16 “standard” and 16 “difficult”, spanning all 4 categories (survival, harvesting, combat, and tech tree). We rely on our Minecraft knowledge to decide the difficulty level. “Standard” tasks require fewer steps and lower in-game resource dependencies to complete.
> * 32 creative tasks: 16 “standard” and 16 “difficult”. Tasks labeled with “standard” are relatively shorter-horizon tasks.
>
> For all tasks above, we recommend that researchers run 100 test episodes each and report the percentage success rate. The programmatic tasks have ground-truth success, while the creative tasks need MineCLIP evaluation. We added a new section in the main paper (Sec. 3.3) to discuss the starter tasks and evaluation protocol.  We have also open-sourced MineCLIP, model weights, and agent code for the community to build upon: https://github.com/MineDojo/MineCLIP
>
> > _*"While there is a focus in the paper and dataset on open-ended learning, a key aspect of open-ended embodied AI is generalisation, and this doesn't seem to be discussed much or tackled by the benchmark as-is. [...] it would be great to leverage that to enable evaluating how general the learned policies are."*_
>
> Great point! We have added a set of new experiments (Sec. 6) to test the zero-shot visual generalization capability of the learned agents. Given the same task, our policy is only trained in a single, canonical environment setting: plain terrain + sunny + noon. At test time, we evaluate the policy without finetuning on a combination of unseen weather (rainy, snowy, and thunder), lighting conditions (noon, sunset, and midnight), and terrains (desert, forest, and ice plain) — 27 scenarios in total.
>
> For the baseline, the agents use the original OpenAI-CLIP image encoder (not trained on our YouTube videos) and their policy heads are trained by behavior cloning on successful trajectories. Then we evaluate them zero-shot on the same 27 test scenarios. The robustness against visual shift can be quantitatively measured by the relative performance degradation on novel test scenarios for each task. In the table below, numbers outside parentheses are percentage success rates of 200 test episodes (averaged over 3 random seeds), while those inside parentheses are relative performance changes.
>
> | Tasks | Ours (attn), train env | Ours (attn), unseen test env | OpenAI CLIP, train env | OpenAI CLIP, unseen test env |
> |:---:|:---:|:---:|:---:|:---:|
> | Milk Cow | 64.5 +- 37.1 | 64.8 +- 31.3 (+0.8%) | 90.0 +- 0.4 | 29.2 +- 3.7 (-67.6%) |
> | Hunt Cow | 83.5 +- 7.1 | 55.9 +- 7.2 (-32.9%) | 72.7 +- 3.5 | 16.7 +- 1.6 (-77.0%) |
> | Combat Spider | 80.5 +- 13.0 | 62.1 +- 29.7 (-22.9%) | 79.5 +- 2.5 | 54.2 +- 9.6 (-31.8%) |
> | Combat Zombie | 47.3 +- 10.6 | 39.9 +- 25.3 (-15.4%) | 50.2 +- 7.5 | 30.8 +- 14.4 (-38.6%) |
>
> The results show that while all methods incur performance drops, agents with MineCLIP encoder is more robust to visual changes than the baseline across all tasks. Pre-training on diverse in-the-wild YouTube videos is important to boosting zero-shot visual generalization capability, a finding consistent with prior literature [1,2].
>
> * [1] Learning transferable visual models from natural language supervision. Radford et al. 2021.
> * [2] R3M: A Universal Visual Representation for Robot Manipulation. Nair et al. 2022.
>
> **<CONTINUED>**

---

### Official Review · Reviewer_SPDY · 2022-07-13
**MineDojo: Building Open-Ended Embodied Agents with Internet-Scale Knowledge**

**Rating:** 9
**Confidence:** 4
**Correctness:** The dataset and simulator are constru…
**Clarity:** The paper is well written with clarity.

**Strengths:**

This work helps to address a significant problem in existing research in embodied AI research which is the constraint of training in isolated worlds/environments/rooms with limited complexity and diversity. The work aims to build a generalist embodied agent by providing 1) an open-world environment that supports an unlimited variety of open-ended goals and tasks, 2) curating a large-scale database of prior knowledge of this open-world environment that is useful for learning and 3) having a flexible and scalable agent's architecture that allows the conversion of large-scale knowledge into actionable insights. Furthermore, I like the idea of introducing a list of creative tasks that the reward function cannot be manually designed. The proposed novel task evaluation metric for the proposed tasks further shows that these proposed metrics align with human evaluations.

**Weaknesses:**

1. According to line 171, the author describes some of the creative tasks that would be brainstormed. I agree that some of the tasks lack straightforward success criteria, and humans cannot manually define the reward function. However, the tasks under creative tasks in Table 1 use tasks such as "find ocean, find the nether portal, etc" I am not sure if this can be classified as a form of navigation task in which the reward can be defined as long as the state space is made known. So my question is, how does the author truly quantify creative tasks apart from programmatic tasks?
2. How did the paper utilise the unstructured Minecraft wiki information for learning actionable knowledge? This part wasn't mentioned clearly.
3. Just a suggestion, the author could provide a table to show a more straightforward comparison between the different platforms supporting Minecraft for AI research. The paper could also give some motivation and applications for the embodied AI research (citing a few survey papers from the field would be good).



**Additional Feedback:**

No additional feedback.

**Documentation:**

The code and instructions for setting up and reproducing the experiments are well documented, the author even provided the code in google colab for people to try out. Maybe, it would be good to have a small subset of the dataset for researchers to perform rapid prototyping on.

**Ethics:**

No, I believe it's fine.



**Relation To Prior Work:**

To my knowledge, the authors provide a good overview of existing work from embodied AI simulator, Minecraft for AI research and internet-scale multimodal knowledge bases. However, for the related work on embodied AI, the author could have given some motivation and applications for the embodied AI research (citing a few survey papers from the field would be good).

**Summary And Contributions:**

This work provides an excellent open-world platform for embodied AI agents to learn continually in solving multi-tasks. The work contributed in three main ways:
1) Provide an environment API on the Minecraft game to help support the learning of a multitude of tasks and goals in an open world.
2) The paper also provided a large-scale database of multimodal knowledge extracted from public domains to support learning in an open world.
3) The paper also introduces a novel agent learning algorithm that leverages a large pre-trained video-language model to tackle multi-tasks learning as a learned reward function.

---

> ### Author Response · Authors · 2022-08-18
> **Response**
>
> Dear reviewer,
>
> Thank you so much for your detailed review, thoughtful comments, and positive evaluation! We are pleased to know that you think our work addresses a significant problem in existing research in embodied AI, which is the lack of diversity and complexity in the environment. We have added new materials, revised the paper, and answered your questions in turn below.
>
> > _*"My question is, how does the author truly quantify creative tasks apart from programmatic tasks?"*_
>
> Thanks for the question! We define creative tasks as “tasks that do not have well-defined or easily-automated success criteria” (L143). While the task of finding terrains and other structures is semantically well defined, it is difficult to implement a success checker because the simulator does not have the exact location information of these structures given a randomly generated world. In principle, we can make a sweep by querying each chunk of voxels in the world to recognize the terrains, but that would be prohibitively expensive. We have added this clarification in the supplementary (Sec. 8.1).
>
> > _*"How did the paper utilise the unstructured Minecraft wiki information for learning actionable knowledge? This part wasn't mentioned clearly."*_
>
> We are sorry for the confusion. The MineCLIP model does not currently use Wiki and Reddit data. Our goal for building MineCLIP is to provide a simple starter agent for the community to build upon. We do have some ideas on how to use the Wiki database for future works. For instance, wiki pages contain detailed recipes for all crafted objects, which could be provided as input or training data for hierarchical planning methods and policy sketches [1]. Furthermore, one may also apply document understanding models such as LayoutLM [2] and DocFormer [3] to extract multimodal features from Wiki pages of interleaving images and text. We have revised Sec. 4 in the main paper to add the clarification.
>
> * [1] Modular Multitask Reinforcement Learning with Policy Sketches. Andreas et al. 2017.
> * [2] LayoutLM: Pre-training of text and layout for document image understanding. Xu et al. 2019.
> * [3] Docformer: End-to-end transformer for document understanding. Appalaraju et al. 2021.
>
>
> > _*"Just a suggestion, the author could provide a table to show a more straightforward comparison between the different platforms supporting Minecraft for AI research. The paper could also give some motivation and applications for the embodied AI research (citing a few survey papers from the field would be good)."*_
>
> Great suggestion! We have added a table of feature comparison with prior Minecraft AI research frameworks [4-11] to the supplementary (Table 1, Sec. 2).
>
> * [4] MineRL: A large-scale dataset of minecraft demonstrations. Guss et al. 2019.
> * [5] Video pretraining (VPT): Learning to act by watching unlabeled online videos. Baker et al. 2022.
> * [6] The multi-agent reinforcement learning in malmÖ (marlÖ) competition. Perez-Liebana et al. 2019.
> * [7] The Malmo platform for artificial intelligence experimentation. Johnson et al. 2016.
> * [8] CraftAssist: A framework for dialogue-enabled interactive agents. Gray et al. 2019.
> * [9] NeurIPS competition IGLU: Interactive grounded language understanding in a collaborative environment. Kiseleva et al. 2021.
> * [10] EvoCraft: A New Challenge for Open-Endedness. Grbic et al. 2021.
> * [11] Benchmarking the spectrum of agent capabilities. Hafner. 2021.
>
> Following your advice, we have also added citations to a few survey papers [12-15] from embodied AI field in the Related Work section of the main paper:
>
> * [12] A Survey of Embodied AI: From Simulators to Research Tasks. Duan et al. 2022.
> * [13] Rearrangement: A Challenge for Embodied AI. Batra et al. 2020.
> * [14] Recent Advances in Robot Learning from Demonstration. Ravichandar et al. 2020.
> * [15] A Review of Physics Simulators for Robotic Applications. Collins et al. 2021.
>
>
> > _*"It would be good to have a small subset of the dataset for researchers to perform rapid prototyping on."*_
>
> Thanks for the recommendation! For YouTube, we have already provided a tutorial partition (https://doi.org/10.5281/zenodo.6641142) that is a small subset of the entire database. The tutorial partition contains 34K videos and tends to be higher quality than the other videos. We will also curate starter subsets of Wiki and Reddit databases in the future.

---

> > ### Comment · Reviewer_SPDY · 2022-08-19
> > **Reponses to rebuttal**
> >
> > I would like to thank the authors for the detailed response and also taking in my suggestions to make the necessary revision. As a result, i will raise my score to an 9, a good paper needs to be recognize!

---

> > > ### Author Response · Authors · 2022-08-19
> > > **Thank you so much!**
> > >
> > > Dear Reviewer,
> > >
> > > Thank you so much for the update. We truly appreciate your kind words and very constructive feedback!

---

### Official Review · Reviewer_e1eG · 2022-07-21
**Great resource for building general-purpose agents with extensive documentation**

**Rating:** 9
**Confidence:** 4

**Strengths:**

- Extensive evaluation on 12 tasks.
- The dataset contains rich multimodal collections of crowd knowledge.
- The proposed MINECLIP tips to make language-conditioned reward models efficient can be useful for future embodied AI benchmarks.
- GitHub documentation and the gym-style API make this an excellent resource.

**Weaknesses:**

- To my understanding, neither MINECLIP nor the multitask RL agent are open-sourced. This seems to be a necessary component for future works to compare with.
- Would be nice to include GATO and Unified-IO in related work which does not require unified observation and action spaces across tasks. Motivating the need for such unification would also be good to include as reasoning for this new benchmark.
- Please include more details on line 215 (" to preserve the layout information, we additionally save the screenshots of entire pages and extract 5.8M bounding boxes of the visual elements"). A visual example of the extracted elements would be sufficient.



**Additional Feedback:**

- Any plans for open-sourcing the learned models?
- How easy is it for users to extend the framework with more tasks?

**Clarity:**

The paper is generally well-written. The supplementary material contains information for several key details, from descriptions of the observation and action spaces to limitations and societal impact.

**Correctness:**

The evaluation and experimental design seem carefully crafted. The benchmark is also carefully designed and already open-sourced on GitHub with extensive documentation and a clear API. Wondering if the authors are considering open-sourcing the construction process for users to extend this work to other platforms and tasks, and how easy it is to incorporate new tasks.

**Documentation:**

Supplementary material contains lots of details, and the submission also includes a datasheet. The major concern is that the MINECLIP reward model, as well as the trained RL agent, do not seem to be open-sourced when these would be excellent resources for future research.

**Ethics:**

N/A, some societal impacts are described in supplementary

**Relation To Prior Work:**

Clear motivation and contribution over prior benchmarks.

**Summary And Contributions:**

This work introduces a new Minecraft benchmark for developing general-purpose agents, equipped with large-scale multimodal knowledge from Youtube, Reddit, and Wiki. The benchmark suite includes more than 2000 tasks with unified observation and action spaces. Tasks can be categorized into kinds, programmatic tasks that have well-defined rewards and creative tasks that are more open-ended and thus are not associated with well-defined metrics of success. The paper also introduces 1) MINECLIP, a contrastive reward model that aligns video segments with text instructions (subtitles), and is used as a reward function for training RL agents, and 2) tips and tricks to make MINECLIP more efficient to query.

---

> ### Author Response · Authors · 2022-08-18
> **Response**
>
> Dear Reviewer,
>
> We are very grateful for your thoughtful comments and positive feedback! We are delighted to learn that you find our evaluation and experimental design carefully crafted, MineCLIP useful for future embodied AI works, and API documentation extensive. We address each of your concerns in turn below.
>
> > _*"To my understanding, neither MINECLIP nor the multitask RL agent are open-sourced. This seems to be a necessary component for future works to compare with."*_
>
> > _*"Any plans for open-sourcing the learned models?"*_
>
> Absolutely! We have open-sourced MineCLIP, model weights, and RL agent here: https://github.com/MineDojo/MineCLIP. We hope this could be useful to researchers interested in MineDojo. The algorithm implementation may also be generally applicable to other embodied AI problems as well.
>
> > _*"Would be nice to include GATO and Unified-IO in related work which does not require unified observation and action spaces across tasks. Motivating the need for such unification would also be good to include as reasoning for this new benchmark."*_
>
> Thanks for the additional references! We have updated the Related Work section of the paper to add the citations:
>
> GATO and Unified-IO can learn a single model to solve diverse tasks of different I/O interfaces. In Minecraft, human players use the same set of control for all tasks. Therefore we provide a unified observation and action space for ease of development and training.
>
> > _*"Please include more details on line 215 (" to preserve the layout information, we additionally save the screenshots of entire pages and extract 5.8M bounding boxes of the visual elements"). A visual example of the extracted elements would be sufficient."*_
>
> Thanks for the advice! We have added a visualization of the bounding boxes on sample Wiki pages in the supplementary (Fig. 5, Sec. 5.2).
>
> > _*"Wondering if the authors are considering open-sourcing the construction process for users to extend this work to other platforms and tasks, and how easy it is to incorporate new tasks."*_
>
> Great suggestion! We have cleaned up and open-sourced our human labeling UI to curate creative tasks from YouTube videos: https://github.com/MineDojo/TaskCreationUI. The interface is intuitive to use, easy to adapt, and generally applicable to other platforms.

---

### Official Review · Reviewer_bi9J · 2022-07-23
**Nice platform for building open-ended agents**

**Rating:** 8
**Confidence:** 4

**Strengths:**

1. MineDojo provides diverse and rich benchmark environments. Both manual and automatic approaches are adopted to create orders of magnitude larger number of provided environments than previous benchmarks.
2. The benchmark contains interesting creative tasks that rely more on semantic understanding of languages and typically require learning more complicated behaviors.
3. The environments and the processed knowledge base are open-sourced, which can significantly benefit the community on language-conditioned RL, multi-task learning, and open-ended learning.


**Weaknesses:**

The example research project that trains a language-prompted agent to solve a subset of the task suite is relatively weak compared to the great capability of MineDojo in the following aspects:

1. The Wiki pages and Reddit posts are included in the knowledge base, but the conducted experiments have not utilized them.
It could be better to provide some evidence that these two sources of data can benefit learning interesting and complex skills as mentioned in the Introduction.
2. Experiment results are only presented over 12 tasks while MineDojo supports thousands of tasks. It might be better to explain how and why the 12 tasks are selected.
3. The current version is more of a language-conditioned multi-task RL agent, rather than an open-ended agent.
Some potential enhancements could be testing whether the agent can generalize to open-vocabulary commands and trying to train one agent capable of different task groups.

For future projects that develop agents using the creative task suite, it is unclear how to benchmark their performances since the creative task suite does not have any manual reward or code-defined success criterion.
This work shows that the learned MineCLIP reward is a good evaluation metric for the 4 creative tasks, but it remains unclear whether MineCLIP can scale up / generalize to all 1553 tasks.
Also, it will be of great value to suggest evaluation protocols for future research that play with new tasks (either manually constructed or proposed by an open-ended learning agent) outside the current task suites with MineDojo API.


**Additional Feedback:**

Please refer to the previous fields.

**Clarity:**

The paper is generally well organized and easy to follow. Some details deserve more clarification:
The evaluation metric in Table 1 is not explicitly described in the caption or the main paper.
I guess the first 8 tasks are evaluated with the percentage of success and the last 4 creative tasks are evaluated with the predicted success signal thresholded from MineCLIP reward.


**Correctness:**

Most claims are well supported with details and experimental results.
There might be a bit of overclaiming in the learned evaluation protocol since its alignment with human judgment is only reported on 4 different tasks.

**Documentation:**

The benchmark environment and dataset are open-sourced and documented well.


**Ethics:**

The authors have discussed potential ethical concerns due to the leverage of pre-trained foundation models in supplementary materials.


**Relation To Prior Work:**

This work has thoroughly discussed previous attempts for open-ended learning and also particularly benchmarks built with Minecraft.


**Summary And Contributions:**

This work introduces a multi-task open-ended environment in Minecraft accompanied by a large-scale knowledge base to facilitate building generalizable agents.
The task suite goes beyond previous Minecraft benchmarks both for its massive number of tasks and for the language-conditioned creative tasks that are more complex than programmatic tasks.
The authors also demonstrate an example research project built on MineDojo, which first learns a reward proxy from videos in the dataset and then trains a language-conditioned RL agent with this reward to solve both programmatic and creative tasks.

---

> ### Author Response · Authors · 2022-08-18
> **Response (3/3)**
>
> > _*"It will be of great value to suggest evaluation protocols for future research that play with new tasks (either manually constructed or proposed by an open-ended learning agent) outside the current task suites with MineDojo API."*_
>
> Thank you for the comment! For new programmatic tasks, one can implement ground-truth success criteria by using our simulator API to query the game state information (such as [this section](https://docs.minedojo.org/sections/customization/privileged_obs.html) in our API doc). For new creative tasks, one can describe them in natural language and query MineCLIP for binary success classification. However, human judgment is still important and would be preferred if resources permit.
>
> > _*"Some details deserve more clarification: The evaluation metric in Table 1 is not explicitly described in the caption or the main paper. I guess the first 8 tasks are evaluated with the percentage of success, and the last 4 creative tasks are evaluated with the predicted success signal thresholded from MineCLIP reward."*_
>
> Thanks for the suggestion! Your interpretation is indeed correct. We have updated the Table 1 caption in the revised paper.

---

> ### Author Response · Authors · 2022-08-18
> **Response (2/3)**
>
> ### Additional Experiment 1
>
> We have trained a single agent for all 12 tasks. To reduce the computational burden without loss of generality, the agent is trained by behavior cloning on the successful trajectories generated from the self-imitation pipeline (Sec. 5.2). We summarize the result in the table below. Similar to our main experiments, all numbers represent percentage success rates averaged over 3 seeds, each tested for 200 episodes.
>
> Compared to the original agents, the new 12-multitask agent sees a performance boost in 6 tasks, degradation in 4 tasks, and roughly the same success rates in 2 tasks.
> This result suggests that there are both positive and negative task transfers happening. To improve the multi-task performance, more advanced algorithms [7,8] can be employed to mitigate the negative transfer effects.
>
> * [7] Gradient Surgery for Multi-Task Learning. Yu et al. 2020.
> * [8] Understanding and Improving Information Transfer in Multi-Task Learning. Wu et al. 2020.
>
> | Task | Single Agent | Original | Performance Change |
> |:---:|:---:|:---:|:---:|
> | Milk Cow | 91.5 +- 0.7 | 64.5 +- 37.1 | ↑ |
> | Hunt Cow | 46.8 +- 3.7 | 83.5 +- 7.1 | ↓ |
> | Shear Sheep | 73.5 +- 0.8 | 12.1 +- 9.1 | ↑ |
> | Hunt Sheep | 27.0 +- 1.0 | 8.1 +- 4.1 | ↑ |
> | Combat Spider | 72.1 +- 1.3 | 80.5 +- 13.0 | ↓ |
> | Combat Zombie | 27.1 +- 2.7 | 47.3 +- 10.6 | ↓ |
> | Combat Pigman | 6.5 +- 1.2 | 1.6 +- 2.3 | ↑ |
> | Combat Enderman | 0.0 +- 0.0 | 0.0 +- 0.0 | = |
> | Find Nether Portal | 99.1 +- 0.4 | 37.4 +- 40.8 | ↑ |
> | Find Ocean | 95.1 +- 1.5 | 33.4 +- 45.6 | ↑ |
> | Dig Hole | 85.8 +- 1.2 | 91.6 +- 5.9 | ↓ |
> | Lay Carpet | 96.5 +- 0.8 | 97.6 +- 1.9 | = |
>
> ### Additional Experiment 2
> To test the ability to generalize to new open-vocabulary commands, we evaluate the above 12-multitask agent on two novel tasks: “harvest spider string” and “hunt pig”. The table is presented below (also as Table 7, supplementary). Results show that the agent struggles in the zero-shot setting because it has not interacted with pigs or spider strings during training, and thus does not know how to handle them effectively. However, the performance improves substantially by finetuning with the MineCLIP reward. Here the baseline methods are trained from scratch using RL with the MineCLIP encoders and reward. Therefore, the only difference is whether the policy has been pre-trained on the 12 tasks or not. Given the same environment sampling budget, our method significantly outperforms the baseline. It suggests that the multitask agent has learned transferable knowledge on hunting and resource collection, which enables it to quickly adapt to previously unseen tasks.
>
> |  | Hunt Pig | Spider String |
> |:---:|:---:|:---:|
> | Ours (zero-shot) | 1.3 +- 0.6 | 1.6 +- 1.3 |
> | Ours (after RL finetune) | 46.0 +- 15.3 | 36.5 +- 16.9 |
> | Baseline (RL from scratch) | 0.0 +- 0.0 | 12.5 +- 12.7 |
>
> >  _*"For future projects that develop agents using the creative task suite, it is unclear how to benchmark their performances [...] This work shows that the learned MineCLIP reward is a good evaluation metric for the 4 creative tasks, but it remains unclear whether MineCLIP can scale up / generalize to all 1553 tasks."*_
>
> Thanks for the feedback. For creative tasks, we recommend converting MineCLIP into a binary success classifier by following the procedure in Sec. 8.4 of the supplementary. It is also the protocol that we use to evaluate the 4 creative tasks in the paper. For this rebuttal, we have further annotated 50 YouTube video segments each for 5 more tasks that are much more semantically complex, and run MineCLIP evaluation on these videos against a negative set. The agreement with human annotation is measured in the binary classification F1 score below (scaled by 100 for better readability).
>
> | Tasks | Build a farm | Build a fence | Build a house | Ride a minecart | Build a swimming pool |
> |:---:|:---:|:---:|:---:|:---:|:---:|
> | Ours (attn) | 78.7 | 91.4 | 63.7 | 95.9 | 85.0 |
> | Ours (avg) | 73.4 | 83.1 | 37.4 | 96.9 | 94.7 |
>
> We can see that MineCLIP generally has a positive agreement with human judgment, but it is not perfect and does not necessarily work well on all 1,500+ tasks. We note that the current MineCLIP is a proof-of-concept step in leveraging internet data for automated evaluation, and further scaling on more training data and parameters may lead to more improvements. Meanwhile, human judgment remains a useful and important alternative [9,10]. Following your advice, we have toned down the claim in our revised paper (Sec. 6) and added the above results to the supplementary (Sec. 9.3).
>
> * [9] (DALLE-2) Hierarchical Text-Conditional Image Generation with CLIP Latents. Ramesh et al. 2022.
> * [10] (Imagen) Photorealistic Text-to-Image Diffusion Models with Deep Language Understanding. Saharia et al. 2022.
>
> **<CONTINUED>**

---

> ### Author Response · Authors · 2022-08-18
> **Response (1/3)**
>
> Dear Reviewer,
>
> Thank you so much for your positive evaluation, thoughtful comments, and the time taken to provide constructive feedback to strengthen our work further! We are glad to hear that you find our benchmark rich and diverse, our paper well-organized, and open-sourcing efforts beneficial to the research community. We have conducted extra experiments, included more clarifications, and revised the paper to address your concerns.
>
> > _*"The Wiki pages and Reddit posts are included in the knowledge base, but the conducted experiments have not utilized them."*_
>
> Thanks for the question. We acknowledge that MineCLIP does not currently use Wiki and Reddit data (Sec. 4 updated for clarification). Our goal for building MineCLIP is to provide a simple starter agent for the community to build upon. Nonetheless, we have brainstormed on how to use the Wiki and Reddit databases and would like to suggest some potential ideas below. Implementing these ideas will require more effort and computational resources, so they could become full follow-up papers in the future:
>
> 1. The [/r/Minecraft](https://www.reddit.com/r/Minecraft/) subreddit is often functioning as “Stackoverflow for Minecraft,” where people ask questions, and the top-voted comments provide good solutions. Instead of relying on the GPT-3 API, a large language model can be finetuned on the Reddit dataset to generate instructions and execution plans that are better grounded in the Minecraft domain. Concurrent works [1,2,3] have explored similar ideas and showed excellent results on robot learning, which is encouraging for more future research in MineDojo in this direction.
>
> 2. The Wiki database contains detailed recipes for all crafted objects, which could be provided as input or training data for hierarchical planning methods and policy sketches [4]. Furthermore, one may also apply document understanding models such as LayoutLM [5] and DocFormer [6] to extract multimodal features from Wiki pages of interleaving images and text.
>
> * [1] Do As I Can, Not As I Say: Grounding Language in Robotic Affordances. Ahn et al. 2022.
> * [2] Inner Monologue: Embodied Reasoning through Planning with Language Models. Huang et al. 2022.
> * [3] Socratic Models: Composing Zero-Shot Multimodal Reasoning with Language. Zeng et al. 2022.
> * [4] Modular Multitask Reinforcement Learning with Policy Sketches. Andreas et al. 2017.
> * [5] LayoutLM: Pre-training of text and layout for document image understanding. Xu et al. 2019.
> * [6] Docformer: End-to-end transformer for document understanding. Appalaraju et al. 2021.
>
> > _*"Experiment results are only presented over 12 tasks while MineDojo supports thousands of tasks. It might be better to explain how and why the 12 tasks are selected."*_
>
> Great question. We selected these 12 tasks due to the diversity of skills required to solve them (e.g., harvesting, combat, building, navigation) and domain-specific entities (e.g., animals, resources, monsters, terrains, and structures). We can, in principle, scale to many more tasks, but it would require a lot more computational resources with existing RL techniques. We hope the community can build upon the MineCLIP starter agent to design more efficient and scalable multi-task learners.
>
> > _*"The current version is more of a language-conditioned multi-task RL agent, rather than an open-ended agent. Some potential enhancements could be testing whether the agent can generalize to open-vocabulary commands and trying to train one agent capable of different task groups."*_
>
> This is a great suggestion! To address this comment, we have conducted two extra experiments and added the discussion to the supplementary (Sec. 9).
>
> **<CONTINUED>**

---

> ### Comment · Reviewer_bi9J · 2022-08-27
> **Response to authors**
>
> I want to applaud for the great efforts and convincing results during this rebuttal phase and have raised my score to 8.

---

### Official Review · Reviewer_6NLt · 2022-07-25
**Strong contribution towards goal of training general agents**

**Rating:** 9
**Confidence:** 3
**Clarity:** Yes

**Strengths:**

The scale of the dataset provided particularly distinguishes this submission, in addition to the creative leveraging of this dataset to both facilitate training and provide a proposed evaluation method for more open ended tasks. These contributions not only enable research on the specific question of training an versatile language guided agent in complex open-ended environment, but are also relevant for investigation into a variety of topics including pre training for RL and offline learning methods, multitask or meta-learning, etc.  The dataset is broadly accessible and well documented and clear efforts have been made to address the (limited) ethical concerns inherent in collecting a dataset from YouTube and Reddit.

**Weaknesses:**

I see no major weaknesses. To make the proposed automatic evaluation method more compelling, it would be useful to see its performance on the programmatically evaluated tasks in addition to its match to the provided human evaluations on the  4 creative tasks investigated.

**Additional Feedback:**

NA

**Correctness:**

The dataset is constructed in a sound way, with the details clearly documented making for easy reproducibility or modification if desired.  The experiments provided support the claim of potential benefit from both the diversity of available tasks (as seen in the ability to easily craft a set of tasks to evaluate which are diverse but can also be grouped into similar subsets) and the benefit of the provided dataset (as demonstrated by the strong human correlation of the trained reward model and the strong performance of the agent trained using this reward model).

**Documentation:**

The documentation of the dataset collection methods and intended use is thorough and clear.

**Ethics:**

I see no ethical concerns beyond those inherent in datasets collected from YouTube and Reddit (which are addressed by the authors both in the data collection and in the paper).

**Relation To Prior Work:**

The related work section is thorough and clear.

**Summary And Contributions:**


MineDojo includes an extended API for Minecraft, multiple datasets, and a demonstration of one successful path to leverage these components. It provides multiple significant contributions towards the goal of training an agent which can solve a general set of goals in an open-world environment, including (1) a broad set of diverse tasks, both clearly defined and open ended, (2) methods for evaluating the clearly defined tasks along with a proposed, and validated, method for evaluating the open ended tasks (3) an API for extending the task and environment set (4) proof of concept in the form of the training procedure for an agent that can generalize across the set of tasks and (5) a vast environment-specific multimodal knowledge base.

---

> ### Author Response · Authors · 2022-08-18
> **Response**
>
> Dear Reviewer,
>
> Thank you very much for your thoughtful review and positive comments! In particular, we appreciate your kind words regarding the large scale of the database, creative use of the data for agent training, potential to enable new research, and broad accessibility of the framework. We have conducted an extra experiment to address your remaining concern:
>
> > _*"To make the proposed automatic evaluation method more compelling, it would be useful to see its performance on the programmatically evaluated tasks in addition to its match to the provided human evaluations on the 4 creative tasks investigated."*_
>
> We follow the same protocol of creative task evaluation to report performances on the four programmatic tasks, which have ground-truth success criteria. The agreement with ground-truth labels is measured in the binary classification F1 score below, similar to Table 2 (scaled by 100 for better readability). We note that the agreement on Hunt Sheep is lower than the other three programmatic tasks, which likely caused the low RL performance (only 8.1% success rate reported in Table 1). We attribute the low performance of Hunt Sheep to the small volume of training data — hunting sheep for lamb occurs much less frequently than shearing for wool in YouTube videos.
>
> | Task | Milk Cow | Hunt Cow | Shear Sheep | Hunt Sheep |
> |---|---|---|---|---|
> | F1 Score | 99.7 | 88.3 | 99.3 | 64.7 |

---

> > ### Comment · Reviewer_6NLt · 2022-08-28
> > **Response**
> >
> > Thanks for the additional experiments, the results are informative for the (hopefully many) people who will use the platform and build upon your strong baseline results.

---

### Official Review · Reviewer_W4mt · 2022-07-31
**A good environment for open-ended RL research**

**Rating:** 8
**Confidence:** 4
**Clarity:** Yes.

**Strengths:**

Overall, it is a good paper relevant to the broader research community with significant contributions. I especially like the idea of applying a video-test contrastive model to generate reward signals automatically. Potentially, this method could greatly lessen the difficulty of manually designing reward functions for tasks with large knowledge bases.



**Weaknesses:**

I find no major weaknesses in this work.

At the same time, I found the website the authors have built is much more appealing than the paper writing. which the authors could consider to improve their paper presentation.

**Additional Feedback:**

In Table 2, it is shown that MINECLIP consistently agrees well with the ground-truth human judgment, even for MINECLIP(avg). However, there is a large performance gap between using rewards generated with MINECLIP(avg) and using manual rewards, as shown in the first 4 rows in Table 1. Why?
In section 3.1, it is mentioned that the GPT-3-davinci model is used to generate detailed guidance for a subset of the taks. From the presented example, it seems that the generated guidance contains step-by-step information. Is the full guidance used to compute the reward signals or only a part of it is used at each time-step? How useful is the guidance overall? How does a learning agent perform without detailed guidance?


**Correctness:**

Yes.



**Documentation:**

Yes

**Relation To Prior Work:**

The authors discuss related work on four topics: open-ended environments for decision-making agents, Minecraft for AI research, Internet-scale multimodal knowledge bases, and embodied agents with large-scale pre-training. It will be great if the authors can add some comments about a concurrent work(https://openai.com/blog/vpt/) in an updated version.

**Summary And Contributions:**

In this paper, the authors propose a new benchmark for reinforcement learning based on a popular game — Minecraft. It includes thousands of tasks with unified observation and action spaces, and a large knowledge base with multimodal data collected from the Internet. Moreover, a novel learning algorithm is developed to learn reward functions automatically, successfully guiding the learning agent and reducing the burden of designing reward functions manually. In general, it is a good demonstration to show how reinforcement learning could benefit from recent success (e.g., large pre-trained models) in the natural language processes and computer vision.

---

> ### Author Response · Authors · 2022-08-18
> **Response**
>
> Dear Reviewer,
>
> Thank you so much for your thoughtful review and positive evaluation! We are glad that you like the idea of MineCLIP to learn reward functions automatically and lessen the difficulty of manual engineering. We have revised the paper following your suggestions, and will address your remaining questions below.
>
> > _"It will be great if the authors can add some comments about a concurrent work (VPT) in an updated version."_
>
> Absolutely. Thank you for the suggestion. VPT is another great work on learning Minecraft agents. We have updated the Related Work section of the paper to add a discussion of VPT:
>
> VPT [1] is a concurrent work that leverages human contractor data to learn an inverse dynamics model, which can be used to pseudo-label YouTube videos for behavior cloning. VPT learns long-horizon policies effectively, and is complementary to MineCLIP. A pretrained VPT agent can be finetuned to solve language-conditioned open-ended tasks with our learned reward model. Moreover, VPT can be applied to our MineDojo YouTube database for further scaling.
>
> * [1] Video PreTraining (VPT): Learning to Act by Watching Unlabeled Online Videos. Baker et al. 2022.
>
> > _"MineCLIP consistently agrees well with the ground-truth human judgment, but there is a large performance gap between using rewards generated with MineCLIP(avg) and using manual rewards."_
>
> This is an insightful observation. MineCLIP can consistently identify binary task success, but in practice, it can be difficult to be used by RL algorithms. An effective reward function should provide dense and smooth reward signals, giving the agent immediate feedback to accelerate exploration. We hypothesize that MineCLIP[avg] is not as good as MineCLIP[attn] or manual dense reward for providing such smooth and informative reward signals.
>
> > _*"Is the full guidance used to compute the reward signals or only a part of it is used at each time-step? How useful is the guidance overall? How does a learning agent perform without detailed guidance?"*_
>
> Great question. In the current experiments, we have not used detailed guidance -- only the task prompt is used. We have added further clarification to Sec. 3.1 of the revised paper. One idea is to feed each step in the guidance string to MineCLIP sequentially as the agent learns so that it becomes a stagewise reward function for a complex multi-stage task. This is inspired by concurrent works like SayCan [2] and Socratic Models [3]. Doing so may alleviate the dense shaping reward problem mentioned earlier for long-horizon tasks.
>
> * [2] Do As I Can, Not As I Say: Grounding Language in Robotic Affordances. Ahn et al. 2022.
> * [3] Socratic Models: Composing Zero-Shot Multimodal Reasoning with Language. Zeng et al. 2022.

---

### Author Response · Authors · 2022-08-18
**Thank you to all reviewers and meta-reviewers!**

Dear reviewers and meta-reviewers,

We are grateful for all the time you have spent to provide us with constructive feedback and great advice to strengthen our paper even further. We have received six very positive and thoughtful reviews. We appreciate that all reviewers have found our MineDojo framework novel and useful, task suite open-ended and diverse, knowledge bases large-scale and accessible, algorithmic contributions significant, and paper well-organized and clear to read.

For this rebuttal, we have conducted many extra experiments, ablations, and analyses to provide more insight and clarify concerns. We have open-sourced the MineCLIP model code, weights, and creative task labeling tool to facilitate community adoption. In our response to each reviewer below, we address your individual questions and comments. The paper and supplementary PDFs have been updated with suggested revisions, highlighted in yellow. We welcome any follow-up discussions!

---

### Comment · Area_Chair_RRzZ · 2022-08-23
**Please respond to author feedback**

Dear Reviewers,

Please carefully read and respond to all author feedback through the lens of whether it addresses any initial concerns raised.

It is vital for a fair and transparent review process that you read the author’s comments and consider your initial scoring should be adjusted accordingly. If not, please advise the authors why.

Best,

AC

---

### Meta-Review · Area_Chair_RRzZ · 2022-09-03

**Recommendation:** Accept
**Confidence:** 4

**Metareview:**

This paper proposes a new RL benchmark called MineDojo, which consists of thousands of diverse tasks on Minecraft game. The main challenge in designing such a benchmark is to define a good reward function to specify the desired task. This is very tricky since Minecraft is open-ended environment and there are various tasks, where human can't easily define the reward. To address this issue, the authors propose a very interesting idea: utilizing multi-modal (text-video) encoder as a reward. Specifically, the authors pre-trained the multi-modal encoder using contrastive learning (similar to CLIP) using multimodal data collected from the Internet and utilized the similarity between agent's behavior (video) and text as a reward function. Throughout human evaluation, the authors showed that their reward function can induce desired behavior (described by text). Since the authors open-sourced simulator, pre-trained reward model and agents, Minedojo can be a good starting point for making a progress on developing open-ended, multi-task RL agents.

Overall, all reviewers agreed that this is very solid submission and authors also handled concerns from reviewers during discussion period. I recommend acceptance.

---

### Decision · Program_Chairs · 2022-09-16

Accept